# Research on Nickel Material Trade Redistribution Strategy Based on the Maximum Entropy Principle

**DOI:** 10.3390/e24070938

**Published:** 2022-07-06

**Authors:** Xingxing Wang, Anjian Wang, Weiqiong Zhong, Depeng Zhu

**Affiliations:** 1Institute of Mineral Resources, Chinese Academy of Geological Sciences, Beijing 100037, China; xiaoxing0523@126.com; 2Research Center for Strategy of Global Mineral Resources, Chinese Academy of Geological Sciences, Beijing 100037, China; 3School of Geophysics and Information Technology, China University of Geosciences (Beijing), Beijing 100083, China; cugbzdp@163.com

**Keywords:** nickel material, complex network, trade redistribution, maximum entropy principle

## Abstract

In the double carbon background, riding the wind of new energy vehicles and the battery high nickelization, nickel resources rise along with the trend. In recent years, due to the influence of geopolitical conflicts and emergencies, as well as the speculation and control of international capital with its advantages and rules, the world may face price and security supply risks to a certain extent. Therefore, to obtain the most objective trade redistribution strategy, this paper first constructs the nickel material trade network, identifies the core trading countries and the main trade relations of nickel material trade, and finds that the flow of nickel material mainly occurred between a few countries. On this basis, a trade redistribution model is constructed based on the maximum entropy principle. Taking Indonesia, the largest exporter, and the largest trade relationship (Indonesia exports to China) as examples, the nickel material redistribution between countries when different supply risks occur are simulated. The results can provide an important reference for national resource recovery after the risk of the nickel trade.

## 1. Introduction

The core of carbon neutrality is low-carbon and clean energy development. Low-carbon and clean energy is inseparable from the development and application of the new energy industry and electric vehicles. The latter requires the extensive use and development of clean energy technologies and the construction of massive infrastructure and equipment, such as electric cars and solar panels [1]. These equipment need the support of a large number of new energy mineral resources, such as lithium, cobalt, nickel, graphite, and so on [2,3]. It can be predicted that global “carbon neutrality” will greatly increase the demand for new energy minerals in the future. Among them, in the double carbon background, riding the wind of new energy vehicles and the battery high nickelization, nickel resources rise along with the trend [4]. In terms of resource supply, USGS shows that nickel supply is mainly concentrated in Indonesia, Australia, Russia, and other countries, which account for more than 50% of the world’s output.

At present, the contradiction between the supply and demand of nickel resources is increasingly prominent [5]. In recent years, many countries are increasing investment in electric vehicles, and a series of favorable policies for new energy vehicles have emerged frequently, providing unprecedented development opportunities and space for the new energy vehicle industry. Nickel is the main ingredient in the manufacture of electric vehicle batteries. With the development of new energy vehicles, the demand for nickel will also be larger. Some studies show that the demand for nickel will increase by 140–175% by 2025 and 215–350% by 2050 [6]. However, on the one hand, nickel resources are concentrated in a few developing countries and mining policies are unstable [7,8,9,10]; for example, to develop its domestic industry, Indonesia frequently introduces policies to restrict the export of nickel ore. The export restriction of resources is bound to bring certain uncertainties to the security of global resource supply [11]. On the other hand, geopolitical conflicts and emergencies are likely to affect the safe supply of resources. The COVID-19 outbreak in 2020 has put the supply of most key metal minerals at risk of disruption [12]. In March 2022, the outbreak of the Russia–Ukraine conflict caused turmoil in the nickel trading market, which exacerbated the contraction of nickel spot supply in Europe. In short, in addition to the impact of geopolitical conflicts and emergencies, with the promotion of global “carbon neutrality” in the future, the demand and competition for nickel resources related to new energy will be increased, and the international capital will use its advantages and rules to hype and control the nickel resources; the world may face a certain degree of price and security supply risks. So, after the supply risk of nickel resources occurs, how should the nickel resources between countries be reallocated?

In recent years, more and more studies have been made on nickel resources. As early as 2008, Reck et al. (2008) analyzed the man-made nickel cycle of 52 countries or regions in 2000, and the results showed that the nickel cycle was mainly dominated by six countries [13]. Japanese scholars also studied the recovery rate of stainless steel in waste management in Japan by analyzing the material flow of chromium and nickel [14]. Nakajima (2018) discussed the material flow of nickel in global trade among 231 countries and regions [15]. There are many countries in the international nickel trade, and the trade relations between them constitute a complicated trade network. Zhou, et al. [16] used a complex network method to find out the internal evolution law of international nickel ore trade. Yang, et al. [17] helped countries involved in nickel resource trade to find potential trade relations through link prediction methods. However, with the prominent contradiction between supply and demand of nickel resources, the intricate relationships between countries form a trade network. Optimizing the distribution of link weights ensures a more resilient network when supply risks occur in a country or a trade relationship [18]. In the face of ever-present risks, if we want to make the “cake” of global development bigger, we must break down trade barriers and join hands with other countries to achieve mutual benefit. The trade redistribution strategy based on the principle of maximum entropy can precisely help the global trade after risks to achieve this goal [19]. The principle of maximum entropy was put forward by E.T.Jaynes in 1957 [20]. The system risk is analyzed by reconstructing unknown network connections [21,22]. Using the principle of maximum entropy, Ikeda and Iyetomi [23] proposed a model to simulate the structural changes of international trade networks caused by trade tariff changes in the context of government trade policies. Ikeda and Watanabe [24] have developed a model to reconstruct international trade networks by considering the commodity and industrial sectors to study the impact of reducing trade costs. In the game of nickel resource trade, different countries have limited information. Countries participating in the nickel resource trade not only want to make full use of this limited information, but also do not want to make many assumptions about the unknown information, so the maximum entropy method can provide such a probability distribution. The trade distribution obtained in this case is the most objective for the countries involved in the nickel resource trade [25].

Therefore, with the nickel industry chain as the main body, firstly, this paper constructs a global nickel resource trade network. Then, according to the complex network theory, the core trading countries and main trade relations of the nickel resource trade are identified. Finally, a trade redistribution model is built based on the principle of maximum entropy, and different scenarios are set to simulate the nickel resource redistribution among countries when supply risks occur. Section 2 mainly introduces the data and methods, including the construction of the network and the construction of the reconstruction model. Section 3 is the main empirical results. Section 4 provides the main conclusion and discussion.

## 2. Data and Method

### 2.1. Data

According to the life cycle process of the whole nickel industry chain, the nickel-containing products and the nickel content coefficient of nickel-containing products were obtained by referring to the literature of Nakajima, Daigo, Nansai, Matsubae, Takayanagi, Tomita and Matsuno [15]. To obtain more comprehensive trade data, we downloaded the import and export trade data of all products in the nickel industry chain in 2020 from UN Comtrade. However, there may be some inconsistencies between the statistical data reported by a country and its trading partners [26]. For the sake of unification, this paper selects the maximum value in the statistical data of these two countries as the trade value. In addition, to highlight major trading countries and major trade relationships, only the top 95% of trade flows are retained in this paper, and removing these trade flows is unlikely to change the main conclusions and trends, including 93 countries in total.

### 2.2. Construction of the Nickel Material Trade Network

Based on the complex network theory, the nickel material trade network is constructed by the weighting method G = (N, E). In the trade network, N = {1,2,3,…} represents the set of countries involved in trade, E represents the set of trade relations between countries composed of ordered pairs, $e_{ij}$. If i exports nickel resources to j, then $e_{ij}$ = 1; otherwise, $e_{ij}$ = 0. $C_{ij}$ refers to the mass of nickel exported from country i to country j:(1)$$C_{ij}=\sum_{m=1}(W_{ij}^{m}*p_{m})$$
where *m* represents a nickel-containing product in the nickel industry chain,$\;W_{ij}^{m}$ represents the weight of product *m* exported from country i to country j, and $p_{m}$ is the nickel-containing coefficient of product *m*.

Figure 1 shows the international nickel material trade network in 2020. The nodes in the figure represent countries. The larger the nodes are, the more nickel material is exported; the same color of the nodes indicates that countries belong to the same community. The edge represents the trade relationship, the direction of the edge is the direction of the nickel material flow, and the weight of the edge represents the nickel material flow in the trade between the two countries. The wider the edge, the more nickel material flows between the two countries.

### 2.3. Construction of the Nickel Material Redistribution Model

In constructing the nickel material trade redistribution model, this paper argues that, in the short term, each country involved in the nickel resource trade will try to follow the existing trade pattern as closely as possible, because the existing trade pattern may be the best choice for different countries according to their national conditions and trade policies. Therefore, according to the principle of maximum entropy, without any subjective assumptions, the import and export capacity of countries is fixed, and the trade relations between countries are fixed. Based on satisfying the above conditions, a rational resource redistribution strategy is provided for countries participating in the global nickel material trade. In this case, the global nickel resource trade suffered the least loss, and the global trade after redistribution is the most stable.

If a random variable *X* has the value *X* = {$x_{1},x_{2},\ldots x_{k}$}, and its probability distribution is P ($X=x_{i}$) =$\;p_{i}$ (i=1,2,…,n), the entropy of random variable *X* is defined as [27]:(2)$$H\left(X\right)=E[-\ln p_{i}]=-\sum_{i=1}^{n}p_{i}\ln p_{i}$$

Assuming that the total export volume of a country is $E_{i}$ and total import volume is $I_{i}$. On this basis, the bilateral trade volume $w_{ij}$ between countries is used to maximize the configuration entropy *S* [23,28]:(3)$$S=\frac{(\sum_{ij}w_{ij})!}{\prod_{ij}ij!}\approx \left(\sum_{ij}w_{ij}\right)ln\left(\sum_{ij}w_{ij}\right)-\sum_{ij}(w_{ij}lnw_{ij})$$
where $\sum_{ij}w_{ij}$ is a constant; that is, the first half of formula (3) is a constant. Therefore, a convex objective function can be obtained:(4)$$S=-\sum_{ij}(w_{ij}lnw_{ij})$$

Then, entropy *S* is maximized under the following constraints:(5)$$E_{i}=\sum_{j}w_{ij}$$
(6)$$I_{j}=\sum_{i}w_{ij}$$
(7)$$W=\sum_{ij}w_{ij}$$
(8)$$w_{ij}\geq 0$$
where formulas (5)–(7) are constraints; that is, the total import volume, export volume, and total trade volume of each country that participated in the trade are fixed.

However, the trade network has a certain sparsity, which can improve the accuracy of reconstruction. Therefore, to take into account the network sparsity, the objective function (4) is modified by applying the concept of Lasso (Least Absolute Shrinkage and Selection Operator) [29,30]. At this time, our problem can be expressed as the maximization of objective function *g*:(9)$$g(w_{ij})=S-\sum_{ij}w_{ij}{}^{2}=-\sum_{ij}(w_{ij}lnw_{ij})-\beta^{\prime }\sum_{ij}w_{ij}{}^{2}$$
where $\beta^{\prime }$ is a control parameter.

In short, under the condition that the proportion of each country’s exports, imports, and global trade volume in the nickel resource trade network is fixed, the trade relationship between countries is optimized by maximizing $g(w_{ij})$:(10)$$\mathrm{Maximize}\;g(\delta_{ij})=-\sum_{ij}(\delta_{ij}ln\delta_{ij})-\beta^{\prime }\sum_{ij}\delta_{ij}{}^{2}$$
(11)$$\mathrm{Subject}\;\mathrm{to}\;\frac{E_{i}}{W}=\sum_{j}\frac{w_{ij}}{W}=\sum_{j}(\delta_{ij})$$
(12)$$\frac{I_{j}}{W}=\sum_{i}\frac{w_{ij}}{W}=\sum_{i}(\delta_{ij})$$
(13)$$W=\sum_{ij}w_{ij}$$
(14)$$w_{ij}\geq 0$$
where $\delta_{ij}$ can either represent the ratio of country i’s exports to country j to country i’s total exports, or the ratio of country j’s imports from country i to country j’s total imports. $\frac{E_{i}}{W}$ represents the proportion of a country’s total exports to the world’s total trade, $\frac{I_{j}}{W}$ represents the proportion of a country’s total imports to the world’s total trade, and *W* represents the world’s total trade.

## 3. Results and Analysis

### 3.1. Core Trading Countries and Major Trading Relationships

#### 3.1.1. Core Trading Countries

In a network, the weighting degree refers to the sum of weights of all edges connected to a node [31]. In trade networks, it can be used to measure the importance of a country. The weighted indegree ($S_{i}^{in}$) and the weighted outdegree ($S_{i}^{out}$) represent the total imports and exports of a country respectively. It can be calculated by the following formula [32]:(15)$$S_{i}^{in}=\sum_{j=1}^{n}e_{ji}*w_{ji}$$
(16)$$S_{i}^{out}=\sum_{j=1}^{n}e_{ij}*w_{ij}$$
where $w_{ij}$ refers to the weight of edge $e_{ij}$, that is, the total amount of nickel material exported from country i to country j.

In the trade network, betweenness centrality is another indicator to measure the importance of a country, which is used to measure the media capacity of a country in trade. A country’s media capacity can be expressed as:(17)BCi=2n−1(n−2)∑jn∑kngjkigjk, j ≠ k ≠ i, j< k 
where $BC_{i}$ represents the media capability of country i, $g_{jk}$ represents the number of shortest paths between country j and country k, and $g_{jk}\left(i\right)$ represents the number of shortest paths that exist between countries j and country k through country i.

Therefore, this paper lists the main core trading countries in the nickel material trade network from three angles: weighted indegree, weighted outdegree, and media capacity. Table 1 shows that, except For China, Germany, and Japan, the top 10 countries in weighted outdegree are nickel resource producers (orange indicates that the country is a resource producer; green means the country is a net importer of nickel). Indonesia and the Philippines are relatively big exporters, followed by Russia and New Caledonia. The largest weighted indegree is China; while Japan is the second-largest importer, China imports seven times as much. The top countries in media capability are mostly countries with large import capacity, among which China, Japan, and Germany are both big importers and exporters.

#### 3.1.2. Major Trading Relationships

Table 2 shows that 2% of the trade relationships in the nickel material trade network bear approximately 50% of the trade flows, so major trade flows occur between a few countries. The most significant trade flows are from resource-producing countries such as Indonesia, the Philippines, New Caledonia, and Papua New Guinea to China. In addition, nearly half of the trade flows in the top 16 trade relations are from other countries to China, which is highly dependent on other countries for the nickel.

### 3.2. Study on Trade Redistribution of Nickel Resources

To more accurately obtain the trade changes of countries caused by reconstruction, the trade flows between countries under the maximum entropy that are closest to the real trade are obtained by adjusting the control parameter $\beta^{\prime }$ before simulation. In this way, the change of trade volume caused by the maximum entropy of original trade can be reduced. Therefore, after the maximum entropy of the original data (the difference between them is small), the trade flow after the maximum entropy is used as the original data for the trade redistribution analysis under different scenarios. By regression analysis and comparison of the bilateral trade relationship $w_{ij}^{\left(R\right)}$, after redistribution with the real trade relationship $w_{ij}$ under different $\beta^{\prime }$, we find that when $\beta^{\prime }$ is −5, the redistribution of the trade relationship is closest to the true trade value (Figure 2). Therefore, from a global perspective, based on the nickel material redistribution model constructed in Section 2.3, we can obtain the optimal redistribution strategy for global nickel resource trade after supply risks occur in any country or trade relations between countries. This paper takes a typical trading country (Indonesia) and a major trading relationship (Indonesia exports to China) as examples to analyze how global nickel resources can be redistributed to minimize the loss caused by the risk and, thus, make the trade after recovery the most stable.

#### 3.2.1. Trade Redistribution Based on Supply Risk in Core Trading Countries

Nickel is one of the important basic minerals supporting the development of the new energy industry. With the high nickel route becoming an important technical development direction of power batteries, the demand for nickel will also usher in a blowout in the future.

Against this background and trend, global giants are scrambling for nickel resources. Indonesia has the world’s richest laterite nickel ore reserves and production; to accelerate the transformation of Indonesia’s nickel industry, since 2009, Indonesia’s nickel mines ban frequency. The Indonesian government hopes that nickel ore can be processed and smelted locally to add value to the commodity, thus bringing a multiplier effect to its economy. Indonesia’s export restrictions are bound to hit global nickel supplies at a time when the country’s existing processing technology cannot absorb all of its nickel ore. Therefore, we study from a global perspective, how the redistribution of nickel materials in the trade network can stabilize the global trade and minimize the possibility of secondary risks when a supply shock occurs in Indonesia.

This paper takes the 20% reduction of nickel material supply in Indonesia as an example. The analysis shows that when the supply risk occurs in Indonesia, based on the maximum entropy principle, the trade pattern between countries remains unchanged (Figure 3), and the countries involved in nickel material trade are divided into two communities. One community includes Indonesia, the Philippines, New Caledonia, and Papua New Guinea as major resource exporters and China as a major importer. The other is mainly Russia, Canada as the main exporter, importing countries including some European countries, and North American countries.

When the supply of nickel materials in Indonesia is reduced by 20%, according to the principle of maximum entropy, the imports of each country involved in the nickel materials trade will be reduced by 3.5%, and new trade flows between countries can also be obtained. Table 3 lists the top 10 countries in terms of import reduction, and we find that their order is consistent with the weighted indegree ranking in Table 1. At this point, the nickel material trade after the redistribution is the most stable. From a global perspective, it can avoid the confusion of the nickel material trade market caused by the concentration of risk influence.

#### 3.2.2. Trade Redistribution Based on Supply Risk in Major Trade Relationships

China is Indonesia’s largest nickel trading partner, but uncertainties in Indonesia’s export policies, COVID-19, and changes in international trade policies are likely to affect trade flows between countries, leading to a reduction in Indonesia’s nickel exports to China. Then, according to the maximum entropy principle, from a global perspective, without understanding the trade flows between other countries, how can the global nickel material be reconfigured to minimize the risk impact?

This paper takes Indonesia’s supply risk to China as an example. Assuming that Indonesia’s export volume to China decreases by 20% and the reduction will not be exported to other countries, then, according to the maximum entropy principle, the import volume of countries participating in the nickel resource trade will eventually decrease by approximately 2.9%. Analyzing the change in China’s imports, as Indonesia reduces its exports to China, China will increase its imports from countries such as Russia, Canada, Australia, Japan, and Germany. Among them, the increase in China’s imports from Russia accounted for 23% of the decrease in its imports from Indonesia, the increase in imports from Canada accounted for 20% of the decrease, and the increase in imports from Australia and Japan accounted for 15% of the decrease, respectively (Table 4). In addition, in Figure 4, we can also find that Indonesia and China are originally in the same community, but when there is a supply risk from Indonesian exports to China, Indonesia and China are divided into different communities, while Russia, which increases exports to China, are divided in the same community with China. In addition, like Russia and Canada export more to China, they will export less to some European countries, such as the Netherlands, Germany, and Norway. Among them, due to the increase in Canada’s export to China, the decrease in its export to the Netherlands and Germany accounted for approximately 20% of its original export volume, respectively. Similarly, Russia’s exports to the Netherlands and Germany fell by 21% and 16%, respectively (Figure 5).

## 4. Discussion and Suggestions

(1) In the nickel material trade network, 10% of the countries are responsible for more than 70% of the imports and exports of the trade network. Among them, China and Japan have relatively large media capacity and are both big exporters, mainly because these countries have a relatively complete nickel industry chain, importing a large number of primary nickel products, and then exporting a lot of nickel processing products. However, these countries are not resource-producing countries. In order to ensure the stable operation of their own industries, they must ensure the effective supply of upstream nickel resources and avoid the disruption of upstream nickel resource supply affecting the development of the industry. Germany has a large export volume, import volume, and media capacity. On the one hand, it may be related to its large processing capacity. On the other hand, Germany is the main bridge for other European countries to have trade relations with other countries in other regions, importing nickel resources from other regions, and then exporting them to other countries in Europe. Similarly, the United States is the main bridge for countries in the Americas to participate in trade.

(2) As a major global resource producer, other countries are highly dependent on Indonesia for nickel, but its mining policies are extremely volatile. Our research shows that, assuming a 20% reduction in Indonesia’s exports to China and that this reduction does not increase exports to other countries, China will increase its imports from countries such as Russia, Canada, Australia, Japan, and Germany. Among them, the increase in China’s imports from Russia accounted for 23% of the decrease in its imports from Indonesia, the increase in imports from Canada accounted for 20% of the decrease, and the increase in imports from Australia and Japan accounted for 15% of the decrease, respectively. According to the principle of maximum entropy, each country involved in the nickel trade would reduce its imports by approximately 2.9%. However, with the increase of nickel resource demand, it is inevitably more and more difficult to obtain resources. Therefore, on the one hand, resource-demanding countries such as China can reduce external dependence through domestic production, recycling, and substitution, etc. On the other hand, developing a certain national nickel reserve plan is also one of the ways to improve a country’s anti-risk ability.

(3) Global trade patterns are affected by changes in major trade relationships. Indonesia and China used to be in the same community, but when there is a supply risk from Indonesia to China, Indonesia and China were divided into different communities, while Russia, which is exporting more to China, is divided into the same community with China. In addition, like Russia and Canada export more to China, they will export less to some European countries, such as the Netherlands, Germany, and Norway. As Canada increases its exports to China, its exports to the Netherlands and Germany decrease by approximately 20% of its original exports.

(4) From a global perspective, it is in the common interest of all participants to establish international cooperation to stabilize the global supply of nickel resources. On the one hand, due to the impact of resource endowment, most of the world’s nickel materials are dependent on imports, so once the supply of nickel mining countries fluctuates, it will form a big attack on the nickel price of smelting countries. For example, after the Russia–Ukraine conflict broke out in February 2022, driven by multiple factors such as market supply, finance, and trade, nickel prices in the London Metal Exchange (LME) broke the usd100,000/ton mark on 10 March, reaching the highest level in history. LME was forced to suspend nickel trading and declare some transactions invalid. On the other hand, the nickel industry chain of each country is not perfect, and the shortage of nickel resources in the upstream will inevitably lead to the supply risk of nickel-related industries in the downstream along with the industry chain, which will lead to the problem of nickel resource supply security in all countries in the world. Therefore, against the backdrop of the pandemic and rising unilateralism and protectionism, it is all the more important for countries to strengthen trade exchanges and promote rational allocation of resources around the world to achieve mutual benefit and common development.

## 5. Conclusions

Nickel is the standout winner in the decarbonization world. Because nickel has the highest energy density of any combination of metals tested in the positive electrode of a lithium-ion battery, it has become the workhorse of lithium-ion batteries. Coupled with the extreme imbalance and high concentration of nickel resources’ geographical distribution, as well as the uncertainty of supply, the major economies in the world have begun to re-examine and evaluate their supply situation, and formulate the corresponding global resource strategy. Therefore, this paper takes the nickel industry chain as the main body, constructs the global nickel material trade network, and identifies its core trading countries and main trade relations. On this basis, a trade redistribution model is constructed based on the maximum entropy principle. Taking Indonesia, the largest exporter, and the largest trade relationship (Indonesia exports to China) as examples, this paper simulates the nickel material redistribution among countries when supply risks occur under the background of maximum entropy to provide corresponding trade redistribution strategies for countries involved in the nickel resource trade. The main conclusions are as follows: (1) the flow of nickel material occurs mainly between a few countries; (2) maintaining a good trade relationship with each trading country is the main way to ease the supply crisis; (3) shifts in the trading relationships of major trading nations have the potential to alter global trade patterns; and (4) building a new pattern of global nickel trade together hand in hand is the best way for all countries to achieve mutual benefit and a win–win situation. At present, with the development of national globalization, the connection and cooperation between countries connected by international trade is getting closer and closer. While economic globalization promotes economic and cultural exchanges among countries, it also greatly increases the risk that regional influences will become global disasters. A reasonable model can help trading countries formulate corresponding risk response strategies, so we hope that our models and analyses can also be applied to other key mineral areas.

In addition, in the research process, we sum up the nickel materials of all nickel products involved in the nickel industrial chain, without considering the division of labor and differences of different countries at different layers of the industrial chain. In the future, we will consider the differences of different countries at different layers of the industrial chain and design different scenarios to study the reconstruction of nickel industry chain trade under the supply risks in different layers and provide more accurate risk response strategies for countries participating in nickel material trade.

## Figures and Tables

**Figure 1 entropy-24-00938-f001:**
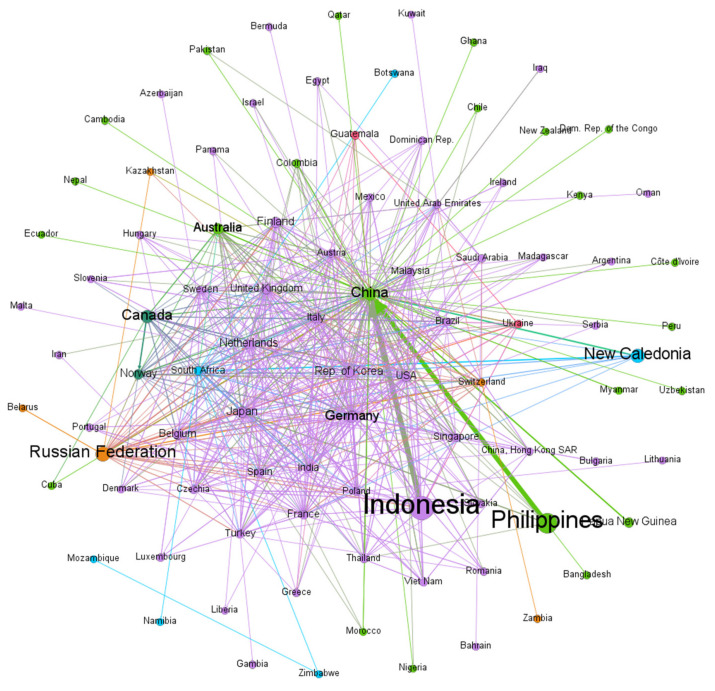
International trade network of nickel material in 2020.

**Figure 2 entropy-24-00938-f002:**
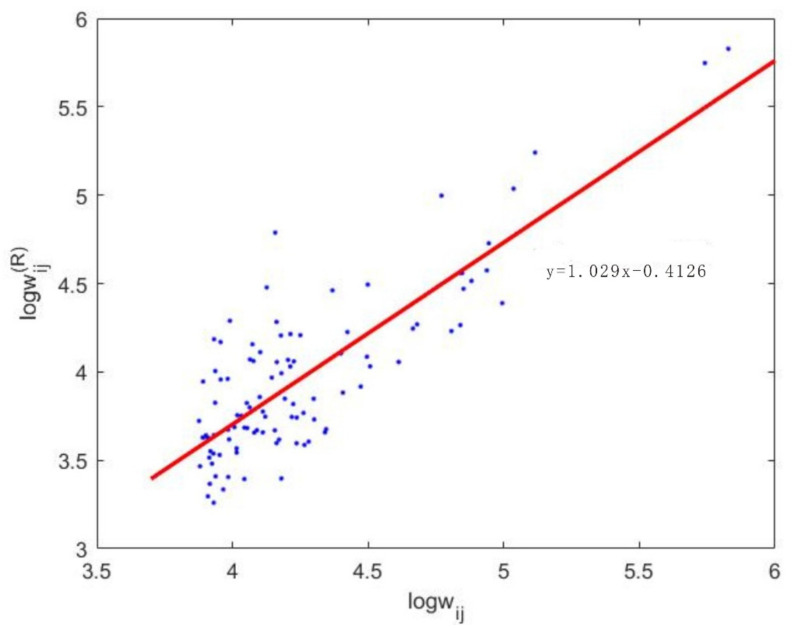
Regression analysis diagram of the redistributed nickel material trade relationship and the actual trade relationship.

**Figure 3 entropy-24-00938-f003:**
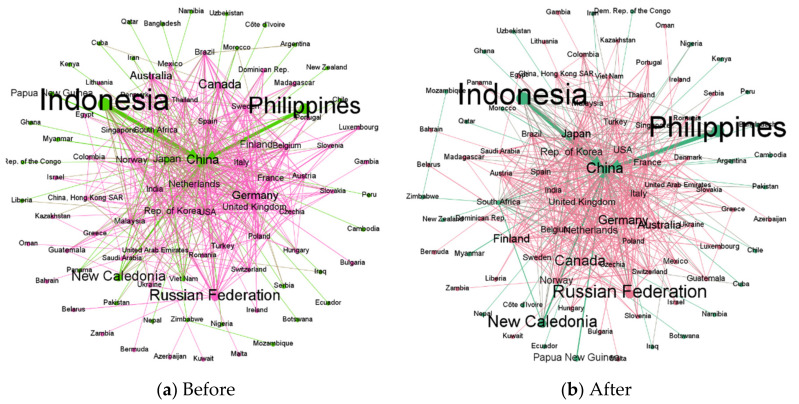
Network comparison of before and after supply risk in Indonesia. Note: (**a**) is the trade network before Indonesia’s supply risk, (**b**) is the trade network after Indonesia’s supply risk. The nodes in the figure represent countries, the size of the nodes represents the volume of exports, and the edges represent the trade relations between countries. The direction of the arrows is consistent with the direction of trade flows, and the same color of the nodes indicates that countries are in the same community.

**Figure 4 entropy-24-00938-f004:**
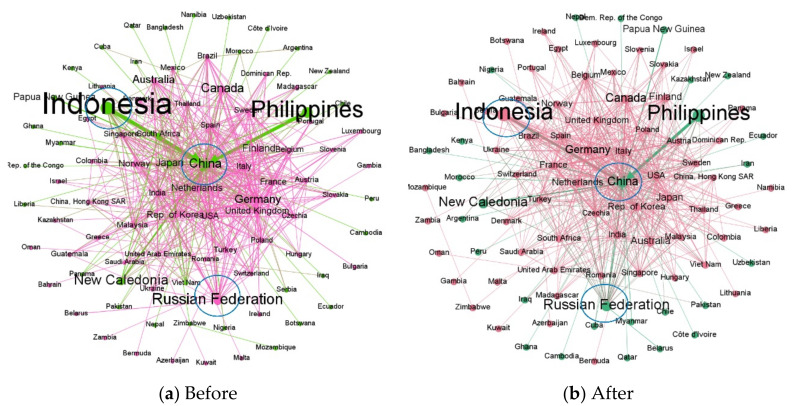
Comparison of nickel material trade networks before and after supply risk (the nodes represent countries, and the different colors of the nodes indicate that each country belongs to different communities. Edges indicate trade relations, and the thickness of edges indicates the size of trade flows. The arrows are in the same direction as trade flows).

**Figure 5 entropy-24-00938-f005:**
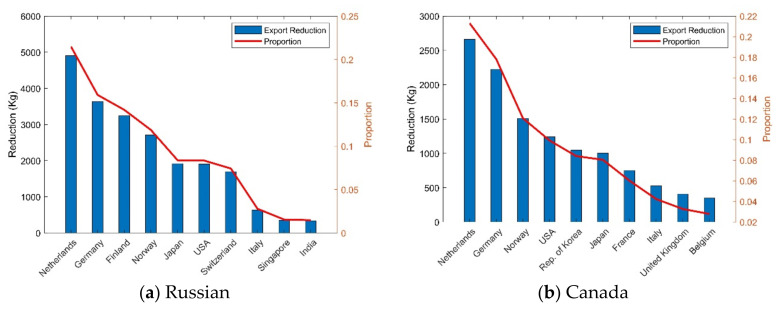
Major export changes for Russia and Canada.

**Table 1 entropy-24-00938-t001:** Top 10 core trading countries.

Rank	Weighted Outdegree	Weighted Indegree	Betweenness Centrality
1	Indonesia 818,934	China 1,941,923	China 1946
2	Philippines 621,484	Japan 258,848	Germany 753
3	Russian Federation 342,539	USA 249,057	Rep. of Korea 277
4	New Caledonia 300,914	Netherlands 239,319	USA 263
5	China 255,811	Germany 224,203	Italy 219
6	Canada 242,607	Rep. of Korea 166,060	South Africa 208
7	Germany 192,295	Norway 124,684	Russian Federation 173
8	Australia 157,638	Italy 110,019	India 172
9	Japan 143,372	France 104,613	Turkey 148
10	Finland 137,616	South Africa 95,372	United Arab Emirates 144
Total	76%	70%	

**Table 2 entropy-24-00938-t002:** Major trade relations in the nickel material trade network.

Rank	Exporter	Importer	Volume (kg)	Proportion
1	Indonesia	China	672,605.72	14.47%
2	Philippines	China	551,208.01	11.86%
3	New Caledonia	China	130,378.71	2.81%
4	Papua New Guinea	China	108,661.21	2.34%
5	Canada	Norway	98,710.67	2.12%
6	New Caledonia	South Africa	88,057.00	1.89%
7	Russian Federation	Netherlands	86,643.86	1.86%
8	Russian Federation	Finland	76,012.82	1.64%
9	Australia	China	71,073.15	1.53%
10	Indonesia	Japan	70,259.72	1.51%
11	Philippines	Japan	69,711.55	1.50%
12	Canada	USA	69,201.12	1.49%
13	Russian Federation	Switzerland	64,111.94	1.38%
14	Russian Federation	China	58,965.61	1.27%
15	Japan	China	47,925.86	1.03%
16	New Caledonia	Rep. of Korea	46,256.12	1.00%
Total	2%		2,309,783.08	49.70%

**Table 3 entropy-24-00938-t003:** Top 10 countries of import reduction due to Indonesia’s supply risk.

Country	V_1_ (kg)	V_2_ (kg)	R (kg)	*p*
China	1,941,922.99	1,873,483.90	68,439.09	0.035
Japan	258,847.71	249,725.15	9122.56	0.035
USA	249,057.42	240,279.90	8777.52	0.035
Netherlands	239,318.79	230,884.49	8434.30	0.035
Germany	224,202.82	216,301.25	7901.57	0.035
Rep. of Korea	166,060.20	160,207.75	5852.45	0.035
Norway	124,684.08	120,289.84	4394.23	0.035
Italy	110,018.68	106,141.30	3877.38	0.035
France	104,613.38	100,926.50	3686.88	0.035
South Africa	95,372.00	92,010.81	3361.19	0.035

Note: V_1_ stands for original import volume, V_2_ stands for redistributed import volume, R refers to the import decrement, and P refers to the ratio of the import decrement to the original import volume.

**Table 4 entropy-24-00938-t004:** Changes in China’s imports.

Exporter	Importer	V_1_ (kg)	V_2_ (kg)	I (kg)	*p*
Russian Federation	China	122,152.88	99,303.59	22849.29	23.01%
Canada	China	73,832.88	61,339.77	12,493.11	20.37%
Australia	China	33,908.21	29,526.37	4381.83	14.84%
Japan	China	21,270.14	18,563.49	2706.65	14.58%
Germany	China	17,988.23	16,046.44	1941.79	12.10%
Netherlands	China	19,902.85	17,795.27	2107.59	11.84%
Finland	China	18,782.39	16,801.55	1980.84	11.79%
Norway	China	15,676.67	14,296.92	1379.75	9.65%
Rep. of Korea	China	8927.88	8249.96	677.92	8.22%
New Caledonia	China	187,617.02	174,122.90	13,494.12	7.75%

Note: V_1_ stands for original import volume, V_2_ stands for redistributed import volume, I refers to the import increment, and P refers to the ratio of the import increment to the original import volume.

## Data Availability

Not applicable.

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
