# Peer review of "Research on Nickel Material Trade Redistribution Strategy Based on the Maximum Entropy Principle"

_entropy, 2022, doi:10.3390/e24070938_

Round 1

Reviewer 1 Report

The paper gives a statistical reconstruction of the nickel trade (both in raw material and consumer goods) as a world wide network. The aim of the paper is to assess the resilience of the trading network with respect to sudden change in the supply or demand. The network is reconstructed using the Maximum entropy principle. The structure of the paper and its mathematical development is based on reference 16 so no new developments are contained in the paper. 

The problemi addressed in the paper is interesting  and the paper could be published provided that its mathematical structure is better explained and one of the main conclusions drawn by the authors (see point 7. below) is better motivated. Here I detail a number of points that should be revised by the authors

1. line 104. The use in the paper of table 1 of HS code is obscure. It is really necessary? in that case it should be better explained 

2. line 107. Why the different countries are numbered as n_1, n_2,...? It should be 1,2,3...... It is easier to say "country  j" instead of "country n_j"

3. line 138. The mathematical model is credited to reference 12 and 16 while the original version is due to A.G. Wilson ( see e.g. A.G. Wilson "The use of entropy maximizing models in the theory of trip distribution. Mode split and route split" in Journal of Transport Economics and Policy, 1969, pp108-126). This paper and other of the same author should be added in bibliography.

4. line 142. The significance of the constraints in formula (5) to (8) is not explained. The lines 147 to 150 should be moved just after formula (8) to explain the nature of the constraints

5. line 146. The lasso technique should be explained. It is related to the free energy in statistical mechanics? Please add references.

6. line 164. What is the meaning of d_{ij}? is the previously introduced e_{ij}?

7. line 192. In the section 3.2, after tuning of the parameter beta to calibrate the model with the real data, a change in the traffic values between two nodes (China and Indonesia) is assumed. Therefore the constraints are changed and the the new values of the nickel traffic on the network are computed by re-maximizing the entropy. However this can not be a forecast of   what would happen in reality, as the authors claims. The maximum entropy method is a principle that allows to select the probability distribution that complies with the constraints representing the available information on the system and has minimal information (i.e. maximal uncertainty or entropy). If relevant information is not considered, the model fails to give the correct values off the traffic between nodes. This seems to be the case since the parameter beta has to be tuned to make the output of the model to coincide with the real data. The same situation could happen after the sudden change in the traffic value between two nodes, therefore we can not confidently take the new maximum entropy traffic values as a trustworthy forecast. The fact that after the sudden change the values of the traffic are close to the previous ones, reflects the fact that the maximum entropy principle select the new minimum informative probability distribution. So this model predicts that the changes in the traffic values are "at least" the ones computed by manent, but nothing prevents that they are much different from the previous ones. So I am skeptical on the predictive power of the method. Al best one can compute a sort of "sensitivity" of the network to changes in the values of the constraints. This issue should be carefully discussed by the authors

Reviewer 2 Report

The manuscript deals with an attractive research topic, having a sound methodology and revealing significant findings in the field of nickel material trade redistribution strategies. However, there is still room for further organizational and argumentative improvements, prior it to be accepted for publication at the Entropy journal. In this respect the following review comments can be considered.

1) At the first two sections authors are recommended to provide a Table in which the global stocks of nickel per country-producer to be quantitatively given. To this end the data of Table 3 can be expanded to top 30 countries, since at its legend it is noted “Top 10 major trade relations”, but the ranking contains 16, no 10, entries. Therefore, the coverage cannot be below the half portion (49.7%), but to cover at least a portion of 75% of all importing-exporting collaborations to be counted for. Therefore, the list expansion towards the top 30 countries it is advisable. Besides, all HS codes of Table 1 have to be accompanied by the physical terminology/titles of the nickel-products’ correlated to. It is not comprehensive for a general readership to be familiarized with these codes, thus, one Table-column can be added by these products’ naming. Moreover all countries of Figure 1 they can be listed in another Table, relating countries with their corresponding quantities. Besides, in case that Table 1 and Figure 1 have been recalled from or retrieved from other published media, then, these literature sources can be denoted at their legends.

2) At the first two sections authors are recommended to organize their theoretical coverage in alignment with the following subsections of nickel history: a) international context, b) Indonesian context, c) chronological overview of the last three decades, showing the key-evolutionary characteristics of nickel i) technology of production, ii) information on trading trends and exporting countries mutually contracted and collaborated.   

3) All input data of equations included at the analysis, they can be accompanied by value-range taken and units’ measured. To this end an extra Table containing all this quantitative information it can be formulated.

4) The values of all data in Table 5  they can be accompanied by units’ measured, accordingly.

5) All text content, including their Tables and Figures, have to be checked and better cross-cited, enabling a better validation and verification of the conducted analysis and research outcomes.

6) At the Discussion section the argumentation of the research outcomes they cannot be conveyed as detached from the environmental consideration on the extraction-production-manufacturing technologies involved. Therefore, all bulleted information can be better organized into short subsections, covering the dimensions: a) technological background, b) environmental considerations, c) bilateral trading agreements and contracts, d) extroversion and internationalization perspectives offered. The critical point here is authors to a) primary approach their findings beyond the indicators-based analysis, towards a more general and descriptive interpretation of results, b) secondly, come to findings of wider/generalized significance, beyond the specific nickel trading. What are those generalized key-aspects in metals’ trading, based but not limited to nickel, accordingly? Such narrative reorganization of the Discussion section can be extended at two cross-cited text pages.  

7) The Conclusions section is missing. At this new section authors can succinctly approach their research findings in terms of: trading constraints, managerial limitations, manufacturing challenges that can be yielded through the conducted analysis and findings. One or two no numerical, no cross-cited, paragraphs they are adequate.

8) The  literature production can be enhanced and enriched with more and relevant published papers, thus, the following indicative list of published papers it can be considered and cited, where matching to the revised manuscript.

Scopus

EXPORT DATE:05 Jun 2022

Yang, Q., Dong, Z., Zhang, Y., Li, M., Liang, Z., Ding, C.

57303853900;36655565600;57215014625;57304603200;57303854000;35208201900;

Who will establish new trade relations? Looking for potential relationship in international nickel trade

(2021) Sustainability (Switzerland), 13 (21), art. no. 11681, .

https://www.scopus.com/inward/record.uri?eid=2-s2.0-85117589498&doi=10.3390%2fsu132111681&partnerID=40&md5=5b6fe93404bbee348b250964630d1e3f

DOI: 10.3390/su132111681

AFFILIATIONS: School of Management, Hebei GEO University, Shijiazhuang, 050031, China;

Research Center of Natural Resources Assets, Hebei GEO University, Shijiazhuang, 050031, China

ABSTRACT: Nickel ore sand and its concentrate are the main sources of raw nickel materials in various countries. Due to its uneven distribution throughout the world, the international trade of nickel ore sand is also unstable. Looking for potential links in the changing international nickel ore trade can help governments find potential partners, make strategic preparations in advance, and quickly find new partners when original trade relationships break down. In this paper, we build an international nickel ore trade network using a link prediction method to find potential trade relations between countries. The results show that China and Italy, China and Denmark, China and Indonesia, and China and India are most likely to establish trade relations within five years. Finally, according to the research results, suggestions regarding the international nickel ore trade are proposed. © 2021 by the authors. Licensee MDPI, Basel, Switzerland.

AUTHOR KEYWORDS: International trade;  Link forecast;  Nickel ore;  Trade relationship forecast

DOCUMENT TYPE: Article

PUBLICATION STAGE: Final

SOURCE: Scopus

Widiatedja, I.G.N.P.

57219903784;

Indonesia’s export ban on nickel ore: Does it violate the world trade organization (wto) rules?

(2021) Journal of World Trade, 55 (4), pp. 667-696. Cited 2 times.

https://www.scopus.com/inward/record.uri?eid=2-s2.0-85110333909&partnerID=40&md5=4ffa9cfc4b49fdb1917eb55b2246a130

AFFILIATIONS: International Law Department, Faculty of Law Udayana University, Bali, Indonesia

ABSTRACT: Many countries have imposed export restrictions based on economic and non-economic objectives. In 2020, Indonesia has followed this trend by imposing an export ban on nickel ore. The European Union reacted by launching a complaint at the WTO, claiming Indonesia’s export ban has unfairly limited its producers’ access to nickel. This article explores the way in which Indonesia could justify banning export on nickel ore, by looking at the current WTO rules and its judicial decisions. This article claims that Indonesia has no justification to impose this ban. Although it will most likely be temporarily applied, and be designed to prevent a critical shortage, nickel is not essential in Indonesia based on its domestic demand, ongoing plans, economic contribution, and the current mining law. Also, the ban will most likely not be justified based on general exception provisions under Article XX of the 1994. © 2021 Kluwer Law International BV, The Netherlands.

AUTHOR KEYWORDS: Article XI GATT;  Article XX GATT;  Export Bans;  Export Restrictions;  General Exception;  Indonesia;  Nickel Ore;  The European Union;  WTO Dispute Settlement;  WTO Rules

DOCUMENT TYPE: Article

PUBLICATION STAGE: Final

SOURCE: Scopus

Taufik, M.J., Martono, D.N., Soelarno, S.W.

57510856900;57212672757;57395851000;

SWOT Analysis in Determining Environmental Risk Management Strategy in Medium Scale Nickel Laterite Mining (Case Study in PT Rohul Energi Indonesia)

(2021) IOP Conference Series: Earth and Environmental Science, 940 (1), art. no. 012023, .

https://www.scopus.com/inward/record.uri?eid=2-s2.0-85122213607&doi=10.1088%2f1755-1315%2f940%2f1%2f012023&partnerID=40&md5=b96fad26b3464ce43c1bf4838a4250ff

DOI: 10.1088/1755-1315/940/1/012023

AFFILIATIONS: School of Environment Science, University of Indonesia, Jakarta, 10430, Indonesia

ABSTRACT: Nickel is an essential metal in modern infrastructure, with significant uses in the stainless-steel industry for less than 65%. Nearly 70% of the world's nickel laterite production comes from Indonesia and the Philippines. The high demand for nickel-based materials globally and the availability of nickel laterite ore in Indonesia make laterite nickel mining in Indonesia inevitable. Medium and small mining companies are more likely to have a more significant impact on the environment than large companies. Therefore, it is necessary to evaluate the implementation of environmental risk and impact management, and develop a company strategy. The research was conducted by qualitative method, namely by descriptive analysis and SWOT analysis. The data was collected in interview text, questionnaire results, field notes, documentation, photos, and videos. SWOT is one of the methods to analyze the strategy of an organization by identifying and measuring strengths and weaknesses, and opportunities and threats of the organization. Result is obtained score 3.77 (the X-axis), and 0.34 (the Y-axis) and will be plotted into a quadrant graph of the SWOT analysis method. The research resulted in PT. REI is in quadrant I. Therefore, the strategy would be utilizing the strengths and opportunities of the organization. © Published under licence by IOP Publishing Ltd.

DOCUMENT TYPE: Conference Paper

PUBLICATION STAGE: Final

SOURCE: Scopus

Hudayana, B., Suharko, Widyanta, A.B.

23670021500;57191203334;57218800759;

Communal violence as a strategy for negotiation: Community responses to nickel mining industry in Central Sulawesi, Indonesia

(2020) Extractive Industries and Society, 7 (4), pp. 1547-1556. Cited 4 times.

https://www.scopus.com/inward/record.uri?eid=2-s2.0-85090300753&doi=10.1016%2fj.exis.2020.08.012&partnerID=40&md5=71acacab5a044b8b4afd55fa5066fd9a

DOI: 10.1016/j.exis.2020.08.012

AFFILIATIONS: Department of Anthropology, Universitas Gadjah Mada, Bulaksumur, Yogyakarta  55281, Indonesia

ABSTRACT: This paper explores why the local community in Bahodopi Sub-District, Morowali District in Central Sulawesi, Indonesia, committed communal violence during conflicts with a nickel mining company. Six acts of communal violence occurred during the ten years of nickel industry operations in the area, driven by the resulting environmental damage, and seizure of residents' land. The research data was sourced from in-depth interviews with six village activists, five district government staff, three police staff, and three company staff, and focus group discussions with twelve participating citizens and 65 citizens who were not active in communal violence. Using social action theory and the structure of political opportunity, the results of this study reveal that village activists are agents that facilitate and advocate for local communities to successfully commit communal violence as a negotiating tool in obtaining compensation for the negative impacts of mining operations. The district government, police and mining company provide opportunities for communal violence to be resolved through negotiations rather than through state justice. To prevent further communal violence, the company must make social investments to build trust and empower the citizens, involve village activists and the local community through CSR and dialog for peaceful negotiations. © 2020 Elsevier Ltd

AUTHOR KEYWORDS: Agencies;  Communal violence;  Local community;  Negotiation;  Nickel company

DOCUMENT TYPE: Article

PUBLICATION STAGE: Final

SOURCE: Scopus

Round 2

Reviewer 1 Report

The authors have addressed my concerns and modified the manuscript. However in some points (see line 188, 189, 1909 they still use n_i instead of i and n_j instead of j as suggested. Please amend the text.

Author Response

Thank you so much for reviewing our paper and we are sorry that we have not completely corrected this kind of problem. Therefore, we have checked the full text and corrected all similar issues. All the revisions are shown in the attachment.

Reviewer 2 Report

At this revised manuscript authors proceeded in an interesting and extended revision of their initial manuscript, having all research parts developed in a systematic manner. The narrative flow is smooth, the methodology is insightful, and the research findings are novel in the field of nickel material trading. In this respect the revised manuscript sustains novel features and it can be accepted for publication at the Entropy journal as is.

Author Response

Thank you so much for your review and affirmation of our manuscript.Wish you all the best to your work.